# Common and Specific Marks of Different Tau Strains Following Intra-Hippocampal Injection of AD, PiD, and GGT Inoculum in hTau Transgenic Mice

**DOI:** 10.3390/ijms232415940

**Published:** 2022-12-14

**Authors:** Isidro Ferrer, Pol Andrés-Benito, Margarita Carmona, José Antonio del Rio

**Affiliations:** 1Department of Pathology and Experimental Therapeutics, University of Barcelona, 08907 L’Hospitalet de Llobregat, Barcelona, Spain; 2Neuropathology Group, Institute of Biomedical Research (IDIBELL), 08907 L’Hospitalet de Llobregat, Barcelona, Spain; 3Network Centre of Biomedical Research of Neurodegenerative Diseases (CIBERNED), Institute of Health Carlos III, 08907 L’Hospitalet de Llobregat, Barcelona, Spain; 4Molecular and Cellular Neurobiotechnology Group, Institute of Bioengineering of Catalonia (IBEC), Barcelona Institute for Science and Technology, Science Park Barcelona (PCB), 08028 Barcelona, Barcelona, Spain; 5Department of Cell Biology, Physiology and Immunology, Faculty of Biology, University of Barcelona, 08028 Barcelona, Barcelona, Spain

**Keywords:** tau, seeding, Alzheimer’s disease (AD), Pick’s disease (PiD), globular glial tauopathy (GGT), hTau, tau splicing

## Abstract

Heterozygous hTau mice were used for the study of tau seeding. These mice express the six human tau isoforms, with a high predominance of 3Rtau over 4Rtau. The following groups were assessed: (i) non-inoculated mice aged 9 months (*n* = 4); (ii) Alzheimer’s Disease (AD)-inoculated mice (*n* = 4); (iii) Globular Glial Tauopathy (GGT)-inoculated mice (*n* = 4); (iv) Pick’s disease (PiD)-inoculated mice (*n* = 4); (v) control-inoculated mice (*n* = 4); and (vi) inoculated with vehicle alone (*n* = 2). AD-inoculated mice showed AT8-immunoreactive neuronal pre-tangles, granular aggregates, and dots in the CA1 region of the hippocampus, dentate gyrus (DG), and hilus, and threads and dots in the ipsilateral corpus callosum. GGT-inoculated mice showed unique or multiple AT8-immunoreactive globular deposits in neurons, occasionally extended to the proximal dendrites. PiD-inoculated mice showed a few loose pre-tangles in the CA1 region, DG, and cerebral cortex near the injection site. Coiled bodies were formed in the corpus callosum in AD-inoculated mice, but GGT-inoculated mice lacked globular glial inclusions. Tau deposits in inoculated mice co-localized active kinases p38-P and SAPK/JNK-P, thus suggesting active phosphorylation of the host tau. Tau deposits were absent in hTau mice inoculated with control homogenates and vehicle alone. Deposits in AD-inoculated hTau mice contained 3Rtau and 4Rtau; those in GGT-inoculated mice were mainly stained with anti-4Rtau antibodies, but a small number of deposits contained 3Rtau. Deposits in PiD-inoculated mice were stained with anti-3Rtau antibodies, but rare neuronal, thread-like, and dot-like deposits showed 4Rtau immunoreactivity. These findings show that tau strains produce different patterns of active neuronal seeding, which also depend on the host tau. Unexpected 3Rtau and 4Rtau deposits after inoculation of homogenates from 4R and 3R tauopathies, respectively, suggests the regulation of exon 10 splicing of the host tau during the process of seeding, thus modulating the plasticity of the cytoskeleton.

## 1. Introduction

Tau can propagate from one neuron to another under certain physiological conditions, and the process is stimulated by neuronal activity [1,2,3]. Tau secretion mainly occurs trans-synaptically, but non-conventional mechanisms may also take place, including vesicular (microvesicles and exosomes) and non-vesicular transport by direct translocation across the plasma membrane [4,5,6,7,8,9,10]. Tau is then internalized via endocytosis, micropinocytosis, and membrane fusion, with the participation of heparin sulfate proteoglycans, LDL receptor-related protein 1 (LRP1), bridging integrator 1 (Bin1), and M1/M3 muscarinic receptors depending on the mechanism [11,12,13,14,15,16,17]. Similar mechanisms are also identified for recombinant tau fibrils, pathological tau in Alzheimer’s disease (AD), and other tauopathies [18,19,20,21,22,23,24,25,26,27,28,29,30,31,32,33,34,35,36,37]. In addition, tau is a component of actin-based nanotubular channels; extracellular tau enhances the formation of nanotubes and facilitates the transfer of tau from one cell to another [38]. Tau may also be secreted in vesicle-free form sensitive to changes in membrane properties, particularly cholesterol and sphingomyelin content [9]. Finally, tau phosphorylation facilitates normal tau secretion and determines the spread and morphology of tau lesions in inoculated mice [7,39].

Tau seeding and spreading in vivo have been assessed following tau inoculation in the brain of mice expressing murine 4Rtau (adult WT mice), transgenic mice expressing murine 3Rtau and 4Rtau, the longest human brain 4Rtau isoform (Alz17 mice), 1N4Rtau containing the P301S mutation (PS19 mice) generated with different promoters, human 3Rtau and 4Rtau at a ratio of 1:1 in a KO murine tau background (6hTau mice), and high levels of human 3Rtau and lower 4Rtau in a KO murine tau background (hTau mice) [40,41,42,43,44,45,46,47,48].

A significant conclusion in the majority of the studies carried out on inoculated mice is that tau seeding and spreading are mainly dependent on the inoculated tau strain, and that tau deposits in the host are reminiscent of those seen in the corresponding human diseases [19,49,50,51,52,53,54,55,56]. However, deposits in oligodendrocytes forming coiled bodies are found following the inoculation of tauopathies, as well as following the inoculation of AD and primary age-related tauopathy (PART) [57]; neuronal deposits are generated following the inoculation of pure cases of aging-related tau astrogliopathy (ARTAG) [58]; cellular deposits are not invariably disease-specific following the inoculation of homogenates from AD and tauopathies, at least in some experiments on WT mice [59,60,61]; and tau seeding and spreading differs in newborn and adult mice [62]. Disparities can be due, in part, to differences between murine and human tau [63,64]. However, disparities can also be related to differences in the ratio and characteristics of the 3Rtau and 4Rtau burden of the host. In a previous study, we used WT and hTau mice inoculated with the same AD inoculum [64]. The current setting analyzes the profiles of tau seeding in hTau mice following the intra-hippocampal inoculation of sarkosyl-insoluble fractions from AD (3R + 4R tauopathy); Pick’s disease (PiD), a prototype 3R tauopathy; and globular glial tauopathy (GGT), a 4R tauopathy with characteristic globular and glial tau inclusions. Our results are compared with those reported for other inoculated transgenic mice expressing different ratios of human or mouse 3Rtau and 4Rtau.

## 2. Results

### 2.1. Characteristics of Human Samples Used for Inoculation

#### 2.1.1. Morphology of Neuronal Phospho-Tau Inclusions

Neurofibrillary tangles, neuronal globular inclusions, and Pick bodies are typical of AD, GGT, and PiD, respectively (Figure 1A). NFTs in AD were immunoreactive with 3Rtau and 4Rtau antibodies, GGT neuronal inclusions were only 4Rtau, and tau inclusions in PiD were only 3Rtau.

#### 2.1.2. Western Blotting of Sarkosyl-Insoluble Fractions Used for Inoculation

Human brain tissue samples were processed in parallel. Sarkosyl-insoluble fractions of AD brain homogenates blotted with anti-tau-P Ser422 showed disease-specific band patterns. Three bands of 68 kDa, 64 kDa, and 60 kDa, and several lower bands of fragmented tau between 50 kDa and 25 kDa, were detected in AD homogenates; lower bands stained with anti-tau-P Ser422 indicated truncated tau at the C-terminal. GGT was characterized by two bands of 68 kDa and 64 kDa, and lower bands of about 50 kDa. Two bands of 64 kDa and 60 kDa, and several strong bands of about 50 kDa, were observed in PiD homogenates. Sarkosyl-insoluble fractions of the control case blotted with the same antibody were negative (Figure 1B). The differences in the density of the bands may reflect differences in the amount of the total tau between AD, GGT, and PiD. However, these differences had no apparent impact on the capacity of seeding, as PiD homogenates have the lowest capacity of seeding (as discussed later).

### 2.2. Characteristics of hTau Mice

#### 2.2.1. WT Mice and Tau Transgenic Mice Expressing Human Tau (hTau)

The hippocampus of WT mice was not stained with anti-3Rtau antibodies except for polymorphic cells of the inner region of the DG, as detailed in a previous study [62]. In contrast, 3Rtau immunoreactivity in the neuropil and neuronal cytoplasm was marked in hTau mice. Neuronal cell bodies and the neuropil of the hippocampus in WT mice showed weak and diffuse 4Rtau immunoreactivity. However, 4Rtau immunoreactivity in hTau mice was moderate in the cytoplasm of neurons but very weak in the neuropil of the hippocampus. The cytoplasm of neurons and glial cells in WT mice and hTau transgenic mice were not stained with the anti-phospho-tau antibody AT8 (Figure 2A). Densitometry further evidenced differences in the expression of 3Rtau and 4Rtau in the CA1 and DG in WT and hTau mice. 3Rtau was significantly increased in hTau mice (*p* < 0.001). In contrast, 4Rtau expression was significantly higher in WT mice (*p* < 0.001) (Figure 2B).

#### 2.2.2. Western Blotting of Total Brain Homogenates in WT and hTau mice

Total brain homogenates were used to identify tau profiles in WT and hTau mice aged 9 months. Western blots were ran in parallel. The antibody Tau 5 showed weak bands of about 68 kDa and 64 kDa in WT mice, and strong bands of about 68 kDa, 64 kDa, and 60–50 kDa, and a weak upper band in hTau mice. The 3Rtau antibody showed two weak bands of about 64 kDa and 60 kDa in WT mice, but several strong overlapping bands between 60 kDa and 75 kDa, and many bands of lower molecular weight, and truncated tau in hTau mice. Weak 4Rtau-immunoreactive bands between 60 kDa and 68 kDa were identified in WT mice, but only one band was of higher molecular weight than the main band in WT mice (Figure 2C).

### 2.3. Inoculation of hTau Mice with Sarkosyl-Insoluble Fractions from AD, GGT, PiD, and Controls

#### 2.3.1. Morphology of Phospho-Tau Deposits in Inoculated hTau Mice

Inoculated mice showed local phagocytes filled with hemosiderin and clear vacuoles at the site of the injection and along the course of the needle, independently of the genotype. Such changes were interpreted as the result of non-specific traumatic injury of the intracerebral injection.

hTau mice unilaterally inoculated in the hippocampus with AD sarkosyl-insoluble fractions at 6 months and euthanized at 9 months showed AT8-immunoreactive deposits in the cytoplasm of neurons in the CA1 region of the hippocampus, DG, and hilus. The morphology of neuronal deposits was reminiscent of pre-tangles, granular aggregates, and dots in the cell processes. NFTs, as those seen in AD, were absent. AT8-immunoreactive threads and dots were also observed in the ipsilateral corpus callosum (Figure 3A–C).

hTau mice unilaterally inoculated in the hippocampus with GGT sarkosyl-insoluble fractions at the same age and euthanized 3 months later showed globular AT8-immunoreactive deposits in the cytoplasm of neurons in the CA1 region of the hippocampus, DG, and hilus. Phospho-tau deposits were unique or multiple, and sometimes extended to the proximal region of the dendrites. Phospho-tau deposits resembled globular neuronal inclusions in GGT. AT8-immunoreactive threads and dots were present in the corpus callosum, although in fewer numbers than in AD-inoculated hTau mice (Figure 3D–L).

hTau mice unilaterally inoculated in the hippocampus with PiD sarkosyl-insoluble fractions showed very few neurons with AT8-immunoreactive deposits in the CA1 region, DG, and cerebral cortex near the site of injection. Neuronal deposits resembled loose pre-tangles. NFTs and Pick bodies were absent (Figure 3M–O).

At the survival times assessed in the present study, no AT8-immunoreactive inclusions were observed in astrocytes or microglia. A few oligodendrocytes with coiled bodies were identified in the corpus callosum following the inoculation of AD homogenates; no globular oligodendroglial inclusions were detected following the inoculation of GGT homogenates. Glial inclusions were absent in PiD-inoculated mice. No tau deposits were seen in mice inoculated with the vehicle alone (Figure 3P). Tau deposits were not produced in animals inoculated with homogenates from control brains (Figure 3Q).

**Figure 3 ijms-23-15940-f003:**
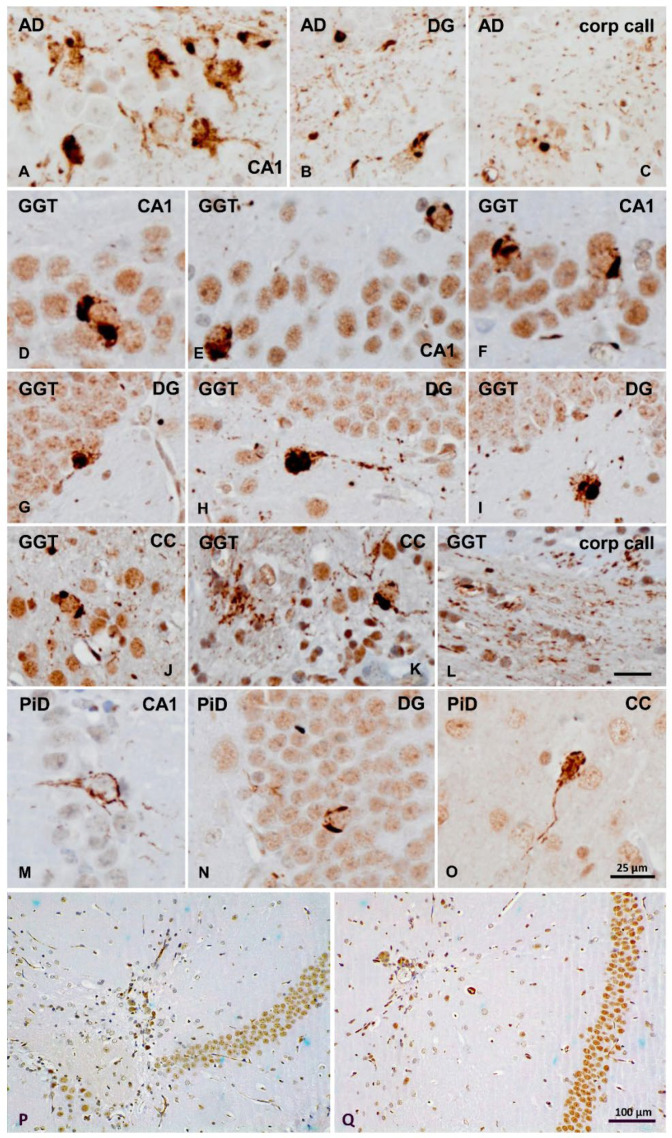
hTau transgenic mice were unilaterally inoculated into the hippocampus with AD (**A**–**C**), GGT (**D**–**L**), and PiD (**M**–**O**) sarkosyl-insoluble fractions at the age of 6 months and euthanized at the age of 9 months. AT8-immunoreactive deposits are depicted in the CA1 region of the hippocampus (CA1), hilus of the dentate gyrus (DG), cerebral cortex (CC), and ipsilateral corpus callosum (corp call). The morphology of deposits differs depending on the disease. AT8-immunoreactive deposits in AD-inoculated mice are pre-tangles, granular aggregates, and dot-like inclusions (**A**,**B**). Phospho-tau neuronal deposits in GGT-inoculated mice are mainly globular, unique, or multiple, sometimes spreading to the proximal dendrites (**D**–**K**). Neuronal inclusions are rare in PiD-inoculated hTau transgenic mice; usually, the inclusions exhibit the morphology of pre-tangles (**M**–**O**). Pick bodies are absent. Phospho-tau-containing threads and dots are also observed in the ipsilateral corpus callosum in AD- and GGT-inoculated, but not in PiD-inoculated, transgenic mice (**C**,**L**). Compare these images with those in Figure 2 corresponding to non-inoculated mice. Phospho-tau-containing threads and dots are not observed in animals inoculated with vehicle alone (**P**) or with PHF-derived from a healthy control (**Q**). Paraffin sections slightly counterstained with hematoxylin; scale bar = 25 µm (**A**–**O**) and scale bar = 100 µm (**P**,**Q**).

Figure 4 presents the distribution of tau deposits in hTau mice unilaterally inoculated in the hippocampus with AD, GGT, PiD; tau deposits were absent in hTau mice inoculated with control homogenates or with vehicle alone (Figure 4). Quantification of AT8 staining over the whole hemisphere shows increased tau levels in animals inoculated with AD, PiD, and GGT compared with mice inoculated with control homogenates or with vehicle alone (Figure 4). In addition, Tau seeding and spreading in the hippocampus was higher in mice inoculated with AD homogenates and lower in PiD-inoculated mice (*p* = 0.01). hTau mice inoculated with GGT homogenates showed intermediate tau levels, when compared with PiD-inoculated mice (*p* = 0.026). GGT-inoculated animals showed a tendency to decreased AT8 levels when compared with AD-inoculated mice (Figure 4).

#### 2.3.2. Co-Localization of Active Tau Kinases and AT8 in Inoculated Mice

Double-labeling immunofluorescence and confocal microscopy of p38-P Thr180/Tyr182 and AT8, and SAPK/JNK-P Thr183/Thr185 and AT8, in hTau mice inoculated with AD and GGT homogenates showed the co-localization of active tau kinases and phospho-tau deposition in neurons (Figure 5). Semi-quantitative studies showed that between 20 and 30% of tau-containing neurons co-localized p38-P and SAPK/JNK-P kinases.

#### 2.3.3. Distribution of 3Rtau- and 4Rtau-Immunoreactive Deposits in Inoculated hTau Mice

hTau transgenic mice inoculated with AD homogenates showed increased cytoplasmic 3Rtau and 4Rtau immunoreactivity in neurons of the CA1 region, DG, and hilus neurons (Figure 6A–C). Tau deposits in GGT-inoculated hTau transgenic mice were mainly stained with anti-4Rtau antibodies (Figure 6D–F), but a small number of deposits were stained with anti-3Rtau antibodies (Figure 6G–I). Tau deposits in hTau mice inoculated with PiD homogenates were stained with anti-3Rtau antibodies, but rare neuronal, thread-like, and dot-like deposits showed 4Rtau immunoreactivity (Figure 6J–L).

Due to differences in the number of total Tau when compared with the number of 3Rtau- and 4Rtau-immunolabeled neurons and threads in GGT and PiD, no attempt was made to assess the densitometric differences of 4Rtau and 3Rtau in inoculated mice.

#### 2.3.4. AD-Tau Inoculation Alters Host MAPT Expression

We assessed the expression of selected molecules linked to exon 10 splicing in the AD-inoculated model showing the most abundant tau seeding and spreading. *MAPT* RNA levels were evaluated in AD-inoculated hTau mice using in situ hybridization. Positive nuclei (pink) were seen in the ipsilateral DG (Figure 7A), but not in the contralateral DG (Figure 7B). In addition, CLK1 protein immunohistochemistry was utilized to assess the expression of CKL1, a mediator of *MAPT* splicing, in the same mice. CKL1 immunoactivity was seen in the hilar cells of the ipsilateral DG (Figure 7C,E). No positive cells were seen in mice inoculated with vehicle alone (Figure 7F). GGT-inoculated animals showed CLK1 immunoreactivity in the ipsilateral corpus callosum (Figure 7G). PiD-inoculated animals did not show CLK1 staining.

## 3. Discussion

Previous studies have shown that the molecular characteristics of tau aggregates differ in AD, GGT, and PiD, including the expression of 3Rtau and 4Rtau isoforms derived from alternative exon 10 splicing, phosphorylation sites, other post-translational modifications, tau conformation, truncation, and aggregation [35,66,67,68,69,70,71,72,73,74,75,76,77,78,79,80]. Cryo-electron microscopy has further demonstrated different tau filament AD, GGT, and PiD conformation [81,82,83]. Different tau strains may explain, in part, the characteristic phenotypes and progression of distinct human tauopathies, and the disease-specific seeding and spreading properties following intracerebral inoculation in mice [30,31,32,84,85,86,87].

In the present study, we have used hTau transgenic mice expressing human 3Rtau and 4Rtau in a KO tau murine background. It is noteworthy that levels of 3Rtau are significantly higher than those of 4Rtau in these transgenic mice [64]. The characteristics of tau deposits in neurons in hTau mice inoculated with sarkosyl-insoluble fractions from AD are similar to those seen in other experimental models using Alz17 Tg, PS19 Tg, WT, and hTau transgenic mice [26,40,49,52,53,54,60,88]. In all these models, neuronal phospho-tau deposits are reminiscent of pre-tangles and tangles.

Neurons with tau deposits in hTau mice inoculated with sarkosyl-insoluble fractions from PiD are scanty compared to those in mice inoculated with AD-tau. Pick bodies are not observed 3 months after inoculating PiD-tau in hTau mice. These findings agree with previous studies showing that PiD sarkosyl-insoluble fractions inoculated in the corpus callosum of WT mice result in relatively limited tau seeding in oligodendrocytes and threads in the ipsilateral and contralateral hippocampus 6 months after injection [57]. PiD-tau has no apparent capacity for seeding in Alz17 Tg mice [49].

Sarkosyl-insoluble fractions from GGT homogenates inoculated in the hippocampus and corpus callosum in WT mice produce phospho-tau seeding and spreading in neurons, threads, and oligodendroglial cells [57,62]. Neuronal deposits are reminiscent of pre-tangles, and inclusions in oligodendrocytes are coiled bodies; however, glial globular inclusions are absent [62]. GGT cases had significantly higher seeding competency when using biosensor cell lines over other tauopathies. Moreover, cellular inclusions in the tau biosensor cell line induced by GGT had a distinct globular morphology [89]. Interestingly, globular neuronal inclusions were also reproduced in our hTau mice using the same inoculum and protocols as those employed in the inoculation of WT mice, in which no globular inclusions were detected [62].

Other mice generated using the CRISPR–Cas9 system express murine 3Rtau and 4Rtau (genome-edited Tau 3R/4R mice) [48]. Sarkosyl-insoluble fractions from AD, PiD, and corticobasal degeneration (CBD) homogenates inoculated in the striatum and hippocampus in these mice develop disease-dependent phospho-tau deposits that spread with time [48]. Neuronal tau deposits resemble pre-tangles and tangles in AD-tau- and CBD-tau-inoculated mice. AT8-positive neurites in the striatum and corpus callosum are found at 8 months and in cell bodies at 12 months following PiD-tau inoculation. Moreover, relatively small, round, Pick-body-like inclusions are identified in the hypothalamus and cerebral cortex 8 months after PiD-tau inoculation [48]. In these particular transgenic mice expressing murine 3Rtau + 4Rtau, neuronal deposits in AD-inoculated mice are 3Rtau and 4Rtau, 4Rtau only in CBD-injected mice, and 3Rtau only in PID-injected mice [48].

In contrast, neuronal inclusions are stained with 3Rtau and 4Rtau antibodies in hTau mice inoculated with PiD-tau. It can be argued that 4Rtau also accumulates in a low percentage of neurons in PiD [90], and this low amount of 3Rtau might be seeded in inoculated mice. However, no 4Rtau inclusions occurred in the PiD case used in the present study. Along the same line, hTau mice inoculated with GGT sarkosyl-insoluble fractions showed 4Rtau and 3Rtau inclusions. However, GGT is a 4R tauopathy; no 3Rtau neuronal and glial inclusions were identified in our GGT case used for inoculation.

Transgenic 6hTau mice express 3R and 4R human tau isoforms in a 1:1 ratio similar to human brains [45]. Inoculation of AD sarkosyl-insoluble fractions produced 3Rtau and 4Rtau neuronal deposits at 3 months and 6 months after inoculation, only 4Rtau deposits following inoculation of progressive supranuclear palsy (PSP) and CBD-tau-enriched fractions, and only 3Rtau deposits in mice inoculated with PiD sarkosyl-insoluble fractions [45]. Inoculation passages in other mice with the material deposited after the first inoculation recapitulate the same 3Rtau and 4Rtau of the primary inoculums [45]. In summary, the characteristics of tau seeds are similar in transgenic mice expressing murine 3Rtau and 4Rtau (genome-edited Tau 3R/4R mice) and human 3Rtau and 4Rtau at a 1:1 ratio (6hTau). However, tau seeds differ in hTau, which express high levels of 3Rtau and low 4Rtau.

These findings support the notion that different tau strains produce different patterns of neuronal tau deposition and seeding, and also that tau seeding depends on the host tau [63,91,92]. Similar conclusions have been recently published regarding α-synucleinopathies. The strain of the initial inoculum dictates the phenotype and pathology of seeding and spreading; however, the burden and spread of inclusion pathology throughout the neuraxis are also host-dependent [93].

A striking fact derived from our experiments is the production of 3Rtau and 4Rtau isoforms after the inoculation of 3R + 4Rtau, 3Rtau, and 4Rtau homogenates obtained from AD, 3R tauopathy (PiD), and 4R tauopathy (GGT). This applies to the present results for hTau mice and those we previously observed in WT mice: 3Rtau and 4Rtau inclusions are also produced following intracerebral inoculation of AD, GGT, ARTAG, and PiD sarkosyl-insoluble fractions in WT adult mice; inoculation of sarkosyl-insoluble fractions from PSP, FTLD-tau, and argyrophilic grain disease (AGD) also show 3Rtau- and 4Rtau-immunoreactive deposits in the host [57,58,60,61,62,91]. Thus, in these models, tau seeding is accompanied by modifications in tau splicing, resulting in the expression of new 3Rtau and 4Rtau isoforms, thus indicating that inoculated tau seeds have the capacity to model exon 10 splicing of the host *mapt* or *MAPT* gene.

Curiously, cerebral ischemia following permanent middle cerebral artery occlusion in rats and mice also induces an increase in 3Rtau versus 4Rtau in oligodendrocytes in the damaged area, and some oligodendrocytic processes show positive staining for 3Rtau [94].

Our preliminary data suggest the presence of *MAPT* gene up-regulation in the ipsilateral hippocampus of AD-inoculated mice. However, mRNA expression may precede, by several days, the translation to protein. Therefore, the present observations call for additional studies and validation using a sequential assessment covering different post-inoculation times.

A shift from fetal to adult tau isoform expression occurs in mice and humans; this is mainly manifested by the progressive decrease in 3Rtau and increase in 4Rtau linked to exon 10 splicing and modulated by the activity of various SR proteins and diverse RNA-interacting and RNA/DNA-binding proteins [95,96,97,98,99]. Six isoforms are expressed in the adult human brain at a ratio of 1:1 for 3Rtau and 4Rtau; but in the adult mouse brain, 4Rtau is predominant [95,100,101]. However, a few neuronal subpopulations in the inner layer of the DG, olfactory bulb, and periventricular regions, and a few neurons in the periventricular hypothalamus, thalamus, basal forebrain, amygdala, and cerebral cortex, contain 3Rtau in adult mice [62].

Modified tau splicing occurs in frontotemporal tauopathies and myotonic dystrophy. Altered tau splicing is commonly associated with mutations in *MAPT* cis-elements or with microsatellite expansion in the non-coding regions of various genes, leading to variations in the expression of splicing factors belonging to the CELF and MBNL families [102,103]. In our AD-inoculated mouse model, the expression of *MAPT*-gene-splicing regulatory kinase protein, CDC-like kinase 1 (CLK1) [104], is increased in selected regions of the ipsilateral hippocampal complex, mainly the hilus of the DG. Kinase deregulation is also observed in the corpus callosum of GGT-inoculated mice, but not in PiD-inoculated hTau mice. However, these observations mark positivity (in the hilus) captured in a single instant, and we have limited knowledge of what occurs in other regions of the hippocampus (CA1 and DG) at earlier times post inoculation.

Functionally, isoforms containing 4Rtau have higher affinity and assembly rates when binding to microtubules (2.5–3.0 times faster) than isoforms containing 3Rtau [95,105,106]. Since the binding of 3Rtau to microtubules is more labile and dynamic, this property may facilitate plasticity during development. Along this line, it can be suggested that the production of 3Rtau following the inoculation of 4Rtau seeds, and 4Rtau the following inoculation of 3Rtau seeds, are active adaptive responses of the host to modulate the plasticity of the cytoskeleton.

## 4. Materials and Methods

### 4.1. Human Brain Samples

Brain samples of the hippocampus were obtained from the Institute of Neuropathology Brain Bank, Bellvitge University Hospital, following the guidelines of the Spanish legislation on this matter (Real Decreto Biobancos 1716/2011) and the approval of the local ethics committee of Bellvitge University Hospital (Hospitalet de Llobregat, Barcelona, Spain). The agonal state was short, with no evidence of acidosis or prolonged hypoxia; the pH values of the brain samples were between 6.8 and 7. At the time of autopsy, one hemisphere was fixed in paraformaldehyde for no less than 3 weeks, and selected brain sections were embedded in paraffin; de-waxed paraffin sections, 4 µm thick, were processed via neuropathological and immunohistochemical methods. The other hemisphere was cut into coronal sections 1 cm thick, and selected brain regions were dissected, immediately frozen at −80 °C, placed in labeled plastic bags, and stored at −80 °C until use; the remaining coronal sections were frozen and stored at −80 °C [107]. The brain samples used in this study were from (i) AD, a 68-year-old man, Braak and Braak NFT stage V–VI, Braak β-amyloid stage C, Thal phase 4, and CERAD 3; (ii) GGT, a man aged 45 years with pyramidal syndrome, cognitive decline, with non-fluent speech, paresis of vertical and horizontal gaze movements, axial rigidity, asymmetric tetraparesis, severe spasticity, and dystonic postures who died at the age of 49 years; the genetic study revealed a P301T mutation in MAPT, and the diagnosis of the post-mortem neuropathological study was GGT type 3 (for details, see case 1 in reference [61]; (iii) PiD, a 67-year-old man with a behavioral variant of frontotemporal dementia and neuropathology of typical Pick’s disease [108]; (iv) one age-matched control case (65-year-old man) with no lesions (Braak 0/0, Thal 0; CERAD 0). All these cases did not show co-morbidities; in particular, argyrophilic grains, thorn-shaped astrocytes, coiled bodies, and other tauopathies were absent. Deposits of α-synuclein and TDP43 were not observed in any region.

### 4.2. Extraction of Sarkosyl-Insoluble Fractions and Western Blotting

Frozen samples of the hippocampus of about 1 g were lysed in 10 volumes (*w*/*v*) with cold suspension buffer (10 mM Tris-HCl, pH 7.4, 0.8 M NaCl, 1 mM EGTA) supplemented with 10% sucrose, protease, and phosphatase inhibitors (Roche). The homogenates were first centrifuged at 20,000× *g* for 20 min (Ultracentrifuge Beckman with 70 Ti rotor), and the supernatant (S1) was saved. The pellet was re-homogenized in five volumes of homogenization buffer and re-centrifuged at 20,000× *g* for 20 min. The two supernatants (S1 + S2) were mixed and incubated with 0.1% *N*-lauroylsarkosynate (sarkosyl) for 1 h while being shaken. Samples were then centrifuged at 100,000× *g* for 1 h. Sarkosyl-insoluble pellets (P3) were re-suspended (0.2 mL/g) in 50 mM Tris-HCl (pH 7.4). Sarkosyl-insoluble fractions were processed for Western blotting, as detailed elsewhere [60].

### 4.3. Animals

The experiments were carried out on heterozygous mice expressing human isoforms of tau protein (hTau; B6.Cg- (GFP)Klt Tg(MAPT)8cPdav/J) in a C57BL/6 background (Jackson laboratories; reference: RRID:IMSR_JAX:005491; Bar Harbor, ME, USA) [41,109]. Homozygous animals are not viable. Transgenic mice were identified by genotyping genomic DNA isolated from tail clips using the polymerase chain reaction conditions indicated by Jackson laboratories. Animals were maintained under standard animal housing conditions in a 12 h dark–light cycle with free access to food and water. All animal procedures were carried out following the guidelines of the European Communities Council Directive 2010/63, and with the approval of the local ethical committee (C.E.E.A: Comity Ètic d’Experimentació Animal; University of Barcelona, Spain; ref. 426/18). Animals were euthanized by decapitation, and their brains were rapidly removed and processed for study. The left cerebral hemisphere was dissected on ice, immediately frozen, and stored at −80 °C until used in biochemical studies. The rest of the brain was fixed in 4% paraformaldehyde, cut into coronal sections, and embedded in paraffin. De-waxed sections were stained with hematoxylin and eosin or processed for immunohistochemistry. The following groups were assessed: (i) non-inoculated mice aged 9 months (*n* = 4); (ii) AD-inoculated mice (*n* = 4); (iii) GGT-inoculated mice (*n* = 4); (iv) PID-inoculated mice (*n* = 4); and (v) control-inoculated mice (*n* = 4). Equal numbers of males and females were used in every group of mice. Two hTau mouse were inoculated with vehicle (50 mM Tris-HCl, pH 7.4).

### 4.4. Inoculation of Sarkosyl-Insoluble from AD, GGT, PiD, and Controls into the Hippocampus of hTau Mice

Mice aged 6 months (four per group) were deeply anesthetized with an intraperitoneal injection of ketamine/xylazine/buprenorphine cocktail and placed in a stereotaxic frame after assuring lack of reflexes. For the intra-hippocampal inoculation, we used a Hamilton syringe (volume 2.0 μL); the coordinates were +1.9 mm inter-aural, −1.4 mm relative to bregma, and +1.8 mm DV from the dural surface [65]. A volume of 1.5 μL was injected at a rate of 0.05 μL/min. Total protein concentrations in inoculated sarkosyl-insoluble fractions were similar in the different groups: 4.17 μg/μL for AD, 4.37 μg/μL for GGT, and 4.20 μg/μL for PiD. The syringe was withdrawn slowly over 10 min to avoid leakage of the inoculum. Following inoculation, the animals were kept in a warm blanket and monitored until they recovered from the anesthesia. Carprofen analgesia was administered immediately after surgery and once a day during the next 2 consecutive days. Animals were housed individually with full access to food and water, and they were euthanized by decapitation at the age of 9 months. The brains were rapidly fixed with 4% paraformaldehyde in phosphate buffer, cut in coronal sections, and embedded in paraffin.

### 4.5. Gel Electrophoresis and Western Blotting of Murine Brain Samples

The brains of four non-inoculated hTau mice and three WT mice were processed for gel electrophoresis and Western blotting, as detailed elsewhere [64]. Frozen samples of the left hemisphere were homogenized in RIPA lysis buffer composed of 50 mM Tris/HCl buffer at pH 7.4 containing 2 mM EDTA, 0.2% Nonidet P-40, 1 mM PMSF, and protease and phosphatase inhibitor cocktail (Roche Molecular Systems, Basel, Switzerland). The homogenates were centrifuged for 20 min at 12,000 rpm. Protein concentration was determined with the BCA method (Thermo-Fisher Scientific, Waltham, MA, USA). Equal amounts of protein (12 µg) for each sample were loaded and separated by electrophoresis on 10% sodium dodecyl sulfate–polyacrylamide gel (SDS-PAGE Invitrogen, Thermo-Fisher), and transferred onto nitrocellulose membranes (Amersham, Buckinghamshire, UK). Non-specific bindings were blocked by incubating 3% albumin in PBS containing 0.2% Tween for 1 h at room temperature. After washing, membranes were incubated overnight at 4 °C with antibodies against different forms of tau protein (Table 1). An antibody against β-actin (Sigma, Barcelona, Spain) was used to control protein loading. Membranes were incubated for 1 h with appropriate HRP-conjugated secondary antibodies (1:3000, Dako, Glostrup, Denmark); the immunoreaction was revealed with a chemiluminescence reagent (ECL, Amersham, Buckinghamshire, UK). Densitometric quantification was performed using ImageLab v4.5.2 software (BioRad, Hercules, CA, USA), using β-actin for normalization.

### 4.6. Immunohistochemistry of Human and Mouse Brains

The hippocampus was selected for study in human cases. The whole hemisphere was assessed in mice. De-waxed sections, 4 µm thick, were processed for immunohistochemistry. The sections were boiled in citrate buffer at pH 6 (20 min) to retrieve tau antigenicity. Endogenous peroxidases were blocked by incubation in 10% methanol–1% H_2_O_2_ solution (15 min), followed by 3% normal horse serum solution. The sections were incubated at 4 °C overnight with one of the primary antibodies listed in Table 1. After incubation with the primary antibody, the sections were incubated with EnVision + system peroxidase (Dako-Agilent, Barcelona, Spain) for 30 min at room temperature. The peroxidase reaction was visualized with diaminobenzidine (DAB) and H_2_O_2_. Control of the immunostaining included omission of the primary antibody; no signal was obtained following incubation with only the secondary antibody. Deposits in neurons and related axonal fibers and threads in the hippocampus in inoculated mice are expressed semi-quantitively using the morphometric approach.

### 4.7. Quantification of Abnormal Deposits Revealed by Immunohistochemistry

Photomicrographs of sections of the hippocampus stained with AT8 antibodies were obtained at a magnification of ×200, covering an area of 0.126 mm^2^, using a DP25 camera adapted to an Olympus BX50 light microscope. The pictures, four areas per region per case in every case, were analyzed using Photoshop software. AD-, GGT-, PiD-, and control-inoculated mice were assessed. The density of tau staining was calculated as the intensity of the diaminobenzidine (DAB) precipitate pigment normalized for the total area, and expressed as a percentage of arbitrary units per area.

### 4.8. Double-Labeling Immunofluorescence and Confocal Microscopy

De-waxed sections, 4 µm thick, were stained with a saturated solution of Sudan black B (Merck, Barcelona, Spain) for 15 min to block autofluorescence of lipofuscin granules present in cell bodies, and then rinsed in 70% ethanol and washed in distilled water. The sections were boiled in citrate buffer to enhance antigenicity and blocked for 30 min at room temperature with 10% fetal bovine serum diluted in PBS. Then, the sections were incubated at 4 °C overnight with combinations of primary antibodies against different proteins (Table 1). After washing, the sections were incubated with Alexa488 or Alexa546 (1:400, Molecular Probes, Eugene, OR, USA) fluorescence secondary antibodies against the related host species. Nuclei were stained with DRAQ5™ (1:2000, Biostatus, Shepshed, UK). After washing, the sections were mounted in Immuno-Fluore mounting medium (ICN Biomedicals, Costa Mesa, CA, USA), sealed, and dried overnight. Sections were examined with a Leica TCS-SL confocal microscope. Regarding the number of neurons co-expressing phospho-p38 and phospho-tau, and phospho-SAPK/JNK and tau, data are expressed as the percentage of tau-positive neurons containing active p38 or active SAPK/JNK from the total number of tau-containing neurons in three non-consecutive sections in every case.

### 4.9. In Situ Hybridization

Detection and location of *MAPT* gene expression was performed using the ViewRNA Tissue Assay Kit (ThermoFisher Scientific, Waltham, MA, USA), an in situ hybridation technique for fresh formalin-fixed paraffin-embedded (FFPE) samples, following the manufacturer’s instructions. De-waxed 4 µm thick sections were hybridized with a probe composed of a 20 bp primary sequence (MAPT probe VA1-10746-VT, ThermoFisher Scientific, Waltham, MA, USA) and a secondary extended sequence serving as a template for hybridization. Other regions of the preamplifier were designed to hybridize to multiple bDNA amplifier molecules that create a branched structure. Finally, alkaline phosphatase (AP)-labeled oligonucleotides, which are complementary to bDNA amplifier sequences, bind to the bDNA molecule by hybridization. In Fast Red substrate, red/pink punctuated precipitates were detected by light bright microscope. Sections were slightly counterstained with Gill’s hematoxylin.

### 4.10. Statistics

Differences between wild-type and hTau animals in Western blot data and staining morphometric data were analyzed with Student’s *t*-test using SPSS software (IBM Corp. Released 2013, IBM-SPSS Statistics for Windows, Version 21.0., Armonk, NY, USA). One-way analysis of variance (ANOVA) followed by Tukey’s post hoc test, using SPSS software, was used to compare the means of three groups for each treatment. Graphic design was performed with GraphPad Prism version 5.01 (La Jolla, CA, USA). Outliers were detected using the GraphPad software QuickCalcs (*p* < 0.05). The data are expressed as mean  ±  SEM. Significant difference levels between AD- and GGT-inoculated hTau and PiD-inoculated hTau were set at * *p* < 0.05 and ** *p* < 0.01.

## Figures and Tables

**Figure 1 ijms-23-15940-f001:**
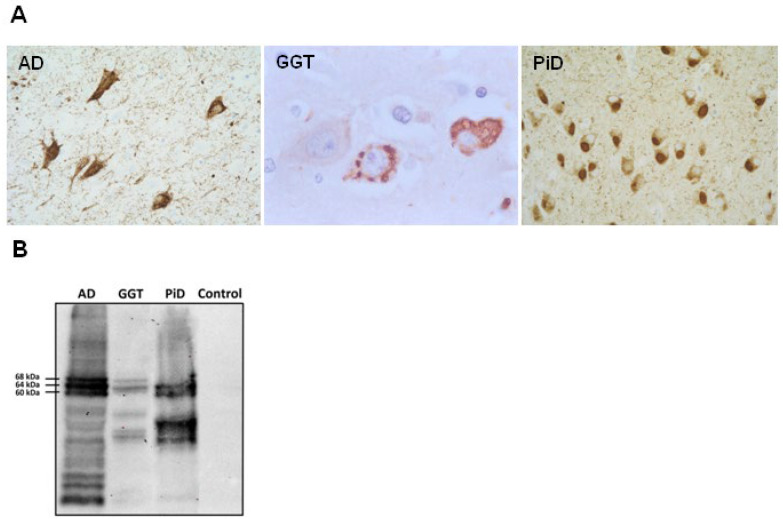
Human case characterization: (**A**) Morphological characteristics of phospho-tau deposits in neurons from AD, GGT, and PiD cases revealed with the AT8 antibody. Neurofibrillary tangles, neuronal globular inclusions, and Pick bodies are typical of AD, GGT, and PiD, respectively. Paraffin sections with slight hematoxylin counterstaining; scale bar = 25 µm. (**B**) Sarkosyl-insoluble fractions of AD brain homogenates blotted with anti-tau-P Ser422 are characterized by three bands of 68 kDa, 64 kDa, and 60 kDa, a weak upper band of 73 kDa, and several lower bands of fragmented tau between 50 kDa and 25 kDa. Lower bands stained with anti-tau-P Ser422 show truncated tau at the C-terminal. GGT blots show two bands of 68 kDa and 64 kDa, and lower diffuse bands of about 50 kDa and 37 kDa. Sarkosyl-insoluble fractions of PiD are characterized by two bands of 64 kDa and 60 kDa and a lower smear. Sarkosyl-insoluble fractions of the control case blotted with the same antibody are negative.

**Figure 2 ijms-23-15940-f002:**
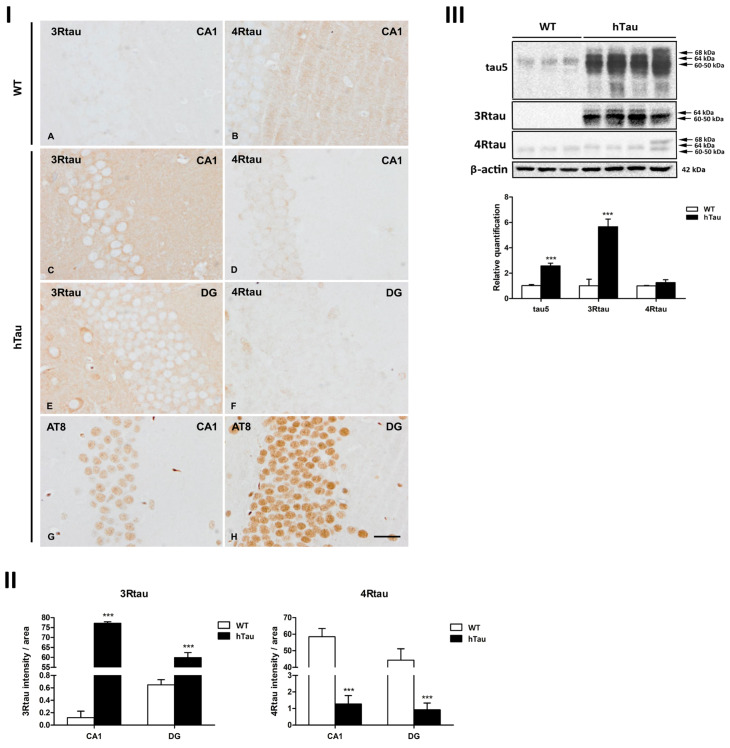
hTau characterization: (**I**) Representative images of 3Rtau (**A**,**C**,**E**), 4Rtau (**B**,**D**,**F**), and AT8 (**G**,**H**) in the CA1 region of the hippocampus and dentate gyrus (DG) in WT (**A**,**B**) and hTau (**C**–**H**) mice at the age of 9 months. Almost absent 3Rtau immunoreactivity in CA1 and weak 4Rtau immunoreactivity occurs in the neuropil in the same region in WT mice (**A**,**B**). In contrast, 3Rtau predominates in the neuropil and cytoplasm of neurons in hTau transgenic mice, whereas 4Rtau antibodies weakly decorate the cytoplasm of CA1 and DG neurons (**C**–**F**). The antibody AT8 stains the nuclei of neurons, but cytoplasmic deposits are absent in hTau transgenic mice. Paraffin sections with slight hematoxylin counterstaining; scale bar = 25 µm. (**II**) Densitometry shows differences in the expression of 3Rtau and 4Rtau in the CA1 and DG in WT and hTau mice. 3Rtau is significantly increased in hTau mice (*p* < 0.001). In contrast, 4Rtau expression is significantly higher in WT mice (*p* < 0.001). (**III**) Gel electrophoresis and Western blotting of total brain homogenates from WT and hTau transgenic mice at 9 months processed with Tau 5, 3Rtau, and 4Rtau antibodies. Antibodies Tau 5 and 4Rtau show weak bands of about 68 kDa and 64 kDa in WT mice; the weak 3Rtau-immunoreactive band may be the expression of a few 3Rtau-immunoreactive region-specific neuronal subpopulations normally found in the adult mouse brain. hTau transgenic mice show Tau 5-immunoreactive bands of 68 kDa, 64 kDa, and 60–50 kDa, accompanied by an upper band. The 3Rtau antibody reveals strong overlapping bands between 60 kDa and 75 kDa, and many bands of lower molecular weight. Weak 4Rtau-immunoreactive bands between 60 kDa and 68 kDa are identified in WT mice, but there was only one band in hTau mice of slightly higher molecular weight than the main band in WT mice in Western blots processed with the same exposure time. The data are expressed as mean ± SEM. Significant difference levels between WT and hTau model were set at *** *p* < 0.001.

**Figure 4 ijms-23-15940-f004:**
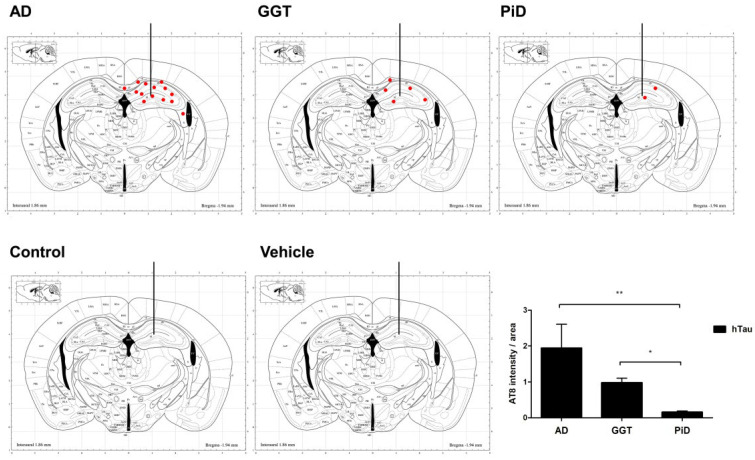
hTau mice inoculated into the hippocampus with AD, GGT, PiD, control sarkosyl-insoluble fractions, and vehicle alone at 6 months and euthanized at 9 months. Tau seeding in the CA1 region and dentate gyrus is higher in mice inoculated with AD homogenates and lower in PiD-inoculated mice. hTau mice inoculated with GGT homogenates show intermediate tau seeding. Tau deposits are absent in hTau mice inoculated with control homogenates and vehicle alone. Vertical line represents the site of inoculation. Diagrams adapted from the mouse brain atlas of Paxinos and Franklin (2019) at Bregma −1.94 coordinate [65]. Dots represent the distribution and relative amount of tau deposits. Graphs show tau densitometry in the hippocampus of inoculated animals. Significant differences between PiD-inoculated hTau mice and AD-, GGT-inoculated hTau mice are set at * *p* < 0.05, ** *p* < 0.01.

**Figure 5 ijms-23-15940-f005:**
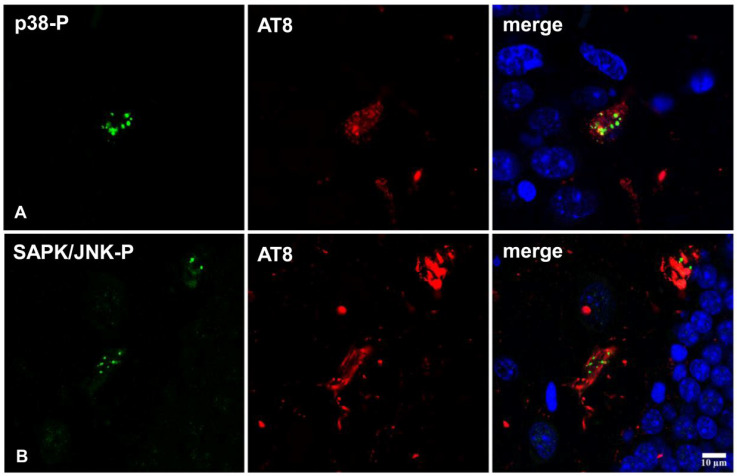
Double-labeling immunofluorescence and confocal microscopy of p38-P Thr180/Tyr182 or SAPK/JNK-P Thr183/Thr185 (green) and AT8 (red) in GGT-inoculated hTau mice at the age of 6 months and euthanized at the age of 9 months. Images show the co-localization of the active kinases with phospho-tau deposition for p38-P (**A**) and SAPK-JNK-P (**B**) in dentate gyrus neurons. Paraffin sections, nuclei are stained with DRAQ5TM (blue); scale bar = 10 µm.

**Figure 6 ijms-23-15940-f006:**
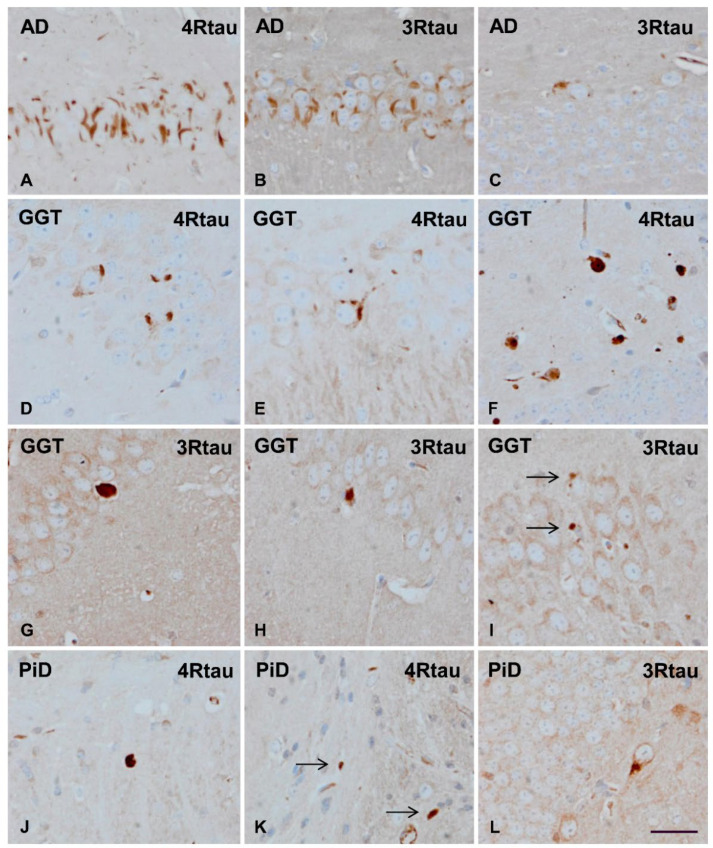
Effect of different sarkosyl-insoluble fractions inoculated in hTau animals: hTau transgenic mice were unilaterally inoculated into the hippocampus with AD (**A**–**C**), GGT (**D**–**I**), and PiD (**J**–**L**) sarkosyl-insoluble fractions at the age of 6 months and euthanized at the age of 9 months. hTau transgenic mice inoculated with AD homogenates show increased 4Rtau (**A**) and 3Rtau (**B**) immunoreactivity in CA1 neurons and dentate gyrus (**C**). Tau deposits in GGT-inoculated hTau transgenic mice are stained with anti-4Rtau antibodies (**D**–**F**) and anti-3Rtau antibodies (**G**–**I**). Tau deposits in hTau mice inoculated with PiD homogenates are stained with anti-4Rtau and anti-3Rtau antibodies (**J**–**L**). In addition to neuronal deposits, tau-immunoreactive dots also occur in the neuropil (thin arrows). Paraffin sections slightly counterstained with hematoxylin; scale bar = 25 µm.

**Figure 7 ijms-23-15940-f007:**
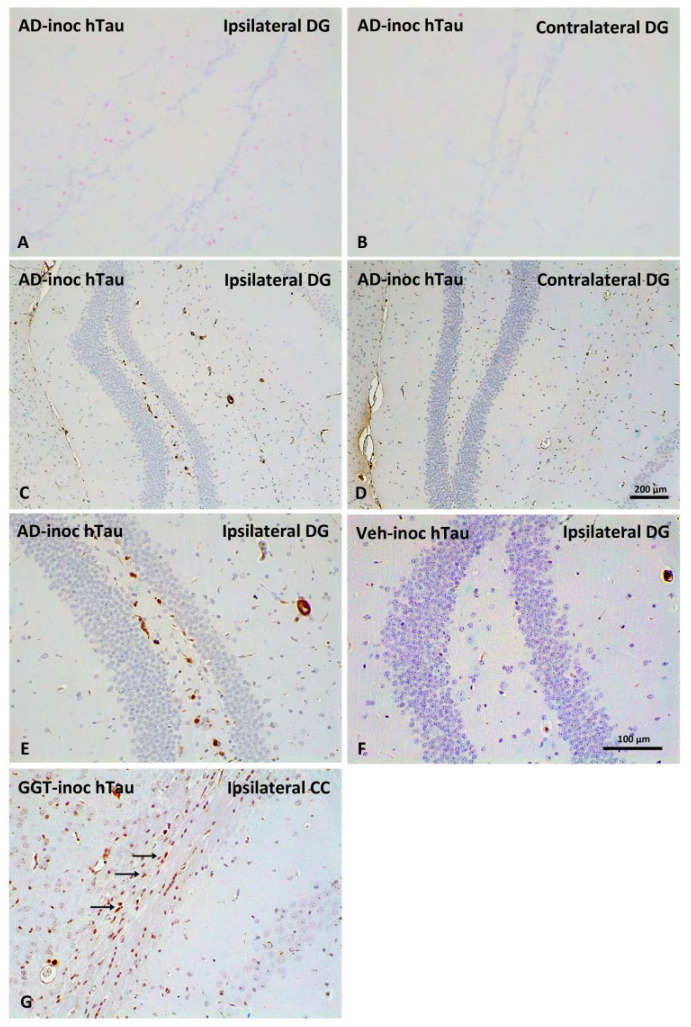
*MAPT* RNA expression and tau splicing modifier CKL1 immunohistochemistry in AD-inoculated hTau mice: hTau transgenic mice were unilaterally inoculated into the hippocampus with AD sarkosyl-insoluble fractions at the age of 6 months and euthanized at the age of 9 months. hTau transgenic mice inoculated with AD homogenates show *MAPT-*positive nuclei (pink), as revealed by in situ hybridization in the ipsilateral dentate gyrus (DG) (**A**), but not in the contralateral DG (**B**). CKL1 protein is expressed in the ipsilateral hilus of the DG of AD-inoculated mice (**C**), but not in the contralateral hilus (**D**); higher magnification of the ipsilateral hilus in AD-inoculated mice (**E**) and vehicle-inoculated mice (**F**). Positive CKL1 immunoreactivity is also observed in the ipsilateral corpus callosum (CC) of GGT-inoculated hTau mice (**G**). Positive staining is detected by punctuate brown precipitates (arrows). Paraffin sections slightly counterstained with Gill’s hematoxylin (**A**,**B**) and Harry’s hematoxylin (**C**–**G**); (**A**–**D**) scale bar 200 µm, (**E**–**G**) scale bar 100 µm.

**Table 1 ijms-23-15940-t001:** List of antibodies used for immunohistochemistry and Western blotting. Abbreviations: 3Rtau, tau with three repeats; 4Rtau, tau with four repeats; p38, p38-kinase; SAPK/JNK, stress-activated protein kinase/Jun amino-terminal kinase; Tau 5, monoclonal recognizing total tau protein; Tau AT8, phosphorylated tau at Ser202/Thr205; Tau-P Ser422, phosphorylated tau at Ser422; WB dil., Western blot dilution; IHQ dil., immunohistochemistry dilution. Suppliers: Upstate, Syracuse, NY, USA; Millipore-Merck, Burlington, MA, USA; Sigma, Barcelona, Spain; Cell Signaling, Danvers, MA, USA; Thermo-Fisher Scientific, Waltham, MA, USA; Innogenetics, Gent, Belgium; LifeSpan Biosciences, Seattle, WA, USA.

Antibody	Supplier	Reference	Host	WB Dil.	IHQ Dil.
3Rtau	Upstate	05-803	Ms	1/1000	1/800
4Rtau	Millipore	05-804	Ms	1/1000	1/50
β-actin	Sigma	A5316	Ms	1/30,000	-
p38-P Thr180/Tyr182	Cell Signaling	9211	Rb	-	1/100
SAPK/JNK-P-Thr183/Thr185	Cell Signaling	9251	Rb	-	1/25
Tau 5	Thermo Scientific	MA5-12808	Ms	1/500	-
Tau AT8-P Ser202/Thr205	Innogenetics	90206	Ms	-	1/50
Tau-P Ser422	Thermo Scientific	44764	Rb	-	1/50
CLK1	LSBio	LS-C382760	Rb	-	1/50

## Data Availability

All the supporting data are in the manuscript.

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
