# Peer review of "Common and Specific Marks of Different Tau Strains Following Intra-Hippocampal Injection of AD, PiD, and GGT Inoculum in hTau Transgenic Mice"

_ijms, 2022, doi:10.3390/ijms232415940_

Round 1
Reviewer 1 Report
The work of Isidro Ferrer and colleagues include the novel data of influence intra-hippocampal injection of AD, PiD, and GGT inoculum in hTau transgenic mice on tau deposits. Authors concluded that the tau strains produce different patterns of active neuronal seeding, which also depend on the host tau. Despite the huge amount of research, several major revisions should be addressed.
1.Females or males mice were used in this study? If both, did you notice the difference? Indicate in the methods.
2.The section « Statistics» is missing. You should all statistics methods transfer in a separate section. What program was used for statistical analysis? Indicate in the methods. Kolmogorov–Smirnov test for determine normality is not applicable for samples of 1-4 animals (The following groups were assessed: i) non-inoculated mice aged 9 months (n=4); ii) AD-inoculated mice (n=4); iii) GGT-inocu-23 lated mice (n=4); iv) PID-inoculated mice (n=4); v) control-inoculated mice (n=4); and vi) inoculated with vehicle alone (n=1)).
3. The conclusion «tau deposits in inoculated mice were expressed semi-quantitatively as +++ for AD, ++ for GGT, + for PiD, and 0 for controls and vehicle alone» is not understandable, line 212-214. How did the semi-quantitative assessment of tau deposits occur? It is better used morphometric analysis of histological slices or western blot analysis for semi-quantitatively assessment, as like on Figure 2 C, and 4 animals will be enough for this.
4.The data of distribution of 3Rtau- and 4Rtau-immunoreactive deposits in inoculated hTau mice (Figure 6) must be supplemented in a semi - quantitative form.
After eliminating the remarks, the article may be recommended for publication in the journal.
Author Response
We thank the reviewers for their useful comments and suggestions. Main changes in the text are highlighted in green.

Reviewer 2 Report
By using Tg/hTau mice, authors determined tau pathology seeded by hippocampal inoculation of sarkosyl-insoluble fractions from AD, Pick's disease (PiD) and globular glial tauopathy (GGT) and found that tau strains from various tauopathies produce different patterns of AT8 immunostaining and 3R-tau and 4R-tau deposits induced by tau seeds from 4R and 3R tauopathies. The study is interesting and can be proved by addressing following comments.
1. Since AD, PiD and GGT display tau pathology with 3R/4R-tau, 3R-tau, and 4R-tau pathology, 3R-tau and 4R-tau in sarkosyl-insoluble fractions should be determined by Western blots.
2. It should be specified that sarkosyl-insoluble fractions were injected in same amounts of tau or protein?
3. Nuclear AT8 immunoactivity was observed in hTau mice in Fig. 2A and some panels of Fig. 3A, suggesting a non-specific immuno-staining.
4. Immunostaining from mice injected with control and vehicle should be included in Figure 3.
5. It will help to verify isoform-specific aggregation by show sarkosyl-insoluble 3R-tau and 4R-tau with Western blots.
6. To conclude the regulation of exon 10 splicing of the host tau during the process of seeding, 3R-tau and 4R-tau should be analyzed by Western blots or RT-PCR.
Author Response
We thank the reviewers for their useful comments and suggestions.
Main changes in the text are highlighted in green.
REVIEWER 2
By using Tg/hTau mice, authors determined tau pathology seeded by hippocampal inoculation of sarkosyl-insoluble fractions from AD, Pick's disease (PiD) and globular glial tauopathy (GGT) and found that tau strains from various tauopathies produce different patterns of AT8 immunostaining and 3R-tau and 4R-tau deposits induced by tau seeds from 4R and 3R tauopathies. The study is interesting and can be proved by addressing following comments.
- Since AD, PiD and GGT display tau pathology with 3R/4R-tau, 3R-tau, and 4R-tau pathology, 3R-tau and 4R-tau in sarkosyl-insoluble fractions should be determined by Western blots.
The band patterns of homogenates used in the present study show the typical pattern of 3R-4Rtau in AD, 4Rtau in GGT and 3Rtau in PiD. This has been validated in numerous studies. Isolated 4Rtau-positive cells can be seen in the neuropathological examination in certain PiD cases, but this is usually not manifested in western blots. Combined forms of AD and PiD have been reported. However, the neuropathological study of our PiD case used for inoculation revealed no 4Rtau pathology in any region. Similarly, a detailed neuropathological study of the GGT case has been published; no 3Rtau pathology was found in this case. PiD and GGT cases are properly referenced in the text in Material and Methods
It should be specified that sarkosyl-insoluble fractions were injected in same amounts of tau or protein?
We have added this relevant information. Approximately, the same amount of total protein was inoculated in each case at concentrations that range from 4.17 to 4.37 μg/μL of total protein. Protein levels of inoculated sarkosyl-insoluble fractions were: 4.17 μg/μL for AD, 4.37 μg/μL for GGT, and 4.20 μg/μL for PiD. These data are added in the Material and Methods section of the revised version.
- Nuclear AT8 immunoactivity was observed in hTau mice in Fig. 2A and some panels of Fig. 3A, suggesting a non-specific immunostaining.
You are right. AT8 immunoreactivity is currently found in the nuclei throughout the murine brain, in contrast to the human brain.
However, we are assessing the presence of AT8 immunoreactivity located in the cytoplasm and cell processes. AT8 immunoreactivity is not found in such structures in physiological conditions but AT8 immunoreactivity is dramatically increased in the cytoplasm and cell processes of neurons, and coiled bodies in transgenic murine models of tauopathies, and in abnormal deposits of tau-inoculated mice.
Other antibodies as Tau-1 and Tau-100 also stain the nuclei of neurons in physiological conditions. The reasons of this antibody-related capacity are not known, but different tau species are localized within the nucleus where they interact with histones and other nuclear proteins.
- Immunostaining from mice injected with control and vehicle should be included in Figure 3.
These images have been included in the figure 3.
- It will help to verify isoform-specific aggregation by show sarkosyl-insoluble 3R-tau and 4R-tau with Western blots.
Due to the small amount of 3Rtau and 4Rtau in the deposits, western blots of total homogenates dilute the signal of 3Rtau and 4Rtau. One possibility could be to carry out western blots of subpopulations of 3R and 4R cells isolated individually under the microscopy. This tremendous task would not add substantial information to that seen by immunohistochemistry.
- To conclude the regulation of exon 10 splicing of the host tau during the process of seeding, 3R-tau and 4R-tau should be analyzed by Western blots or RT-PCR.
We agree in that additional studies should be carried out to explain the modifications in the expression of 3Rtau and 4Rtau following abnormal tau inoculation.
It is worth stressing that adult mice express mainly 4Rtau (in contrast to adult humans which express 4Rtau and 3Rtau). hTau mice express 3Rtau and4Rtau but with a dramatic predominance of 3Rtau. In previous studies properly cited in our manuscript, 3Rtau and 4Rtau is found in the abnormal tau deposits of wild type mice inoculated with AD, PiD, GGT, FTLD, AGD , ARTAG and FTLD-tau homogenates. In the present study, we show exoression of 3R and 4Rtau not only in AD, but also in GGT (4R tauopathy) and PiD (3R taupathy). Therefore, we have several pieces of evidence that exon 10 is modulated following abnormal tau inoculation in WT and hTau mice.
Following your suggestions, we have assessed the expression of CLK1, a modulator of exon 10 splicing. We have seen increased CLK1 immunoreactivity in selected ipsilateral regions of AD- and GGT- inoculated mice. The hilus of the dentate gyrus is currently stained at this particular time-point post-inoculation. An additional experimental approach should be designed to address this important aspect in full by assessing different time-points post-inoculation. We need more than one year to prepare this kind of material.
We have the same problem regarding in situ hybridization. mRNA expression may appear several days or weeks before the translation of the corresponding protein. In the present work, we are studying a unique (advanced) time-point post-inoculation. We have minimal expectancies to observe any change in mRNA expression. Yet, we observed increased MAPT mRNA expression in the ipsilateral hippocampus several months following inoculation. This preliminary result encourages the study of MAPT mRNA expression at earlier time-points post-inoculation.
These preliminary and exciting results are added in the revised version.

Round 2
Reviewer 1 Report
All major concerns have been eliminated by authors. However, minor text editing are required.
Reviewer 2 Report
Although the reviewer is not fully agree, but authors have tried the best to address all comments. They are acceptable.